# Usage of the Brief Job Stress Questionnaire: A Systematic Review of a Comprehensive Job Stress Questionnaire in Japan from 2003 to 2021

**DOI:** 10.3390/ijerph20031814

**Published:** 2023-01-18

**Authors:** Kazuhiro Watanabe, Kotaro Imamura, Hisashi Eguchi, Yui Hidaka, Yu Komase, Asuka Sakuraya, Akiomi Inoue, Yuka Kobayashi, Natsu Sasaki, Kanami Tsuno, Emiko Ando, Hideaki Arima, Hiroki Asaoka, Ayako Hino, Mako Iida, Mai Iwanaga, Reiko Inoue, Yasumasa Otsuka, Akihito Shimazu, Norito Kawakami, Akizumi Tsutsumi

**Affiliations:** 1Department of Public Health, Kitasato University School of Medicine, 1-15-1 Kitazato, Minami-ku, Sagamihara 252-0374, Japan; 2Department of Digital Mental Health, Graduate School of Medicine, The University of Tokyo, 7-3-1 Hongo, Bunkyo-ku, Tokyo 113-0033, Japan; 3Department of Mental Health, Institute of Industrial Ecological Sciences, University of Occupational and Environmental Health, 1-1 Iseigaoka, Yahatanishi-ku, Kitakyushu 807-8555, Japan; 4Department of Mental Health, Graduate School of Medicine, The University of Tokyo, 7-3-1 Hongo, Bunkyo-ku, Tokyo 113-0033, Japan; 5Institutional Research Center, University of Occupational and Environmental Health, 1-1 Iseigaoka, Yahatanishi-ku, Kitakyushu 807-8555, Japan; 6Faculty of Social Policy & Administration, Hosei University, 4342 Aiharamachi, Machida, Tokyo 194-0298, Japan; 7School of Health Innovation, Kanagawa University of Human Services, 3-25-10 Tonomachi, Kawasaki-ku, Kawasaki 210-0821, Japan; 8Institute for Cancer Control, National Cancer Center, 5-1-1 Tsukiji, Chuo-ku, Tokyo 104-0045, Japan; 9Department of Psychiatric Nursing, Graduate School of Medicine, The University of Tokyo, 7-3-1 Hongo, Bunkyo-ku, Tokyo 113-0033, Japan; 10Department of Community Mental Health & Law, National Center of Neurology and Psychiatry, National Institute of Mental Health, 4-1-1 Ogawahigashi, Kodaira, Tokyo 187-0031, Japan; 11Faculty of Human Sciences, University of Tsukuba, 3-29-1 Otsuka, Bunkyo-ku, Tokyo 112-0012, Japan; 12Faculty of Policy Management, Keio University, 5322 Endo, Fujisawa 252-0882, Japan

**Keywords:** questionnaire, psychometric, workplace, job demands, job control, social support

## Abstract

The Brief Job Stress Questionnaire (BJSQ) is used widely in occupational health studies and practice. Summarizing scientific production based on measurement is crucial. This study aimed to systematically review observational studies that used the BJSQ and the New BJSQ to show their usability. A systematic search was conducted for studies investigating relationships between the BJSQ or the New BJSQ subscales and other validated measurements on 13 September 2021, in various literature databases. The BJSQ subscales, scoring methods, and other validated measurements in the studies were qualitatively summarized. In total, 145 published reports between 2003 and 2021 were included. Among the BJSQ subscales, job stressors (n = 95) such as quantitative job overload (n = 65) and job control (n = 64) were most often used. The subscales were utilized to investigate the relationships with several other measurements. Five reports used subscales from the New BJSQ. In the last two decades, the BJSQ and the New BJSQ help measure psychosocial factors (PF) at work and contribute to the publication of scientific papers in the occupational health field. This study would encourage the utilization of the questionnaires for future research and practice.

## 1. Introduction

### 1.1. Background and Previous Work

Conducting multidimensional identification, assessment, and control of psychosocial factors (PF) at work is important in psychosocial risk management for occupational safety and health. Exposure to psychosocial stressors at work leads to physical and mental health problems among workers. Scientific evidence has indicated that high job demands, low job control, low social support, effort–reward imbalance, and high job insecurity elevate the risk of coronary heart disease and mental disorders [1,2,3]. In practice, several countries and regions have guidance or standards, such as the Psychosocial Risk Management—European Framework (PRIMA-EF) [4], the United Kingdom (UK) Health and Safety Executive (HSE) management standards [5], the National Standard of Canada for Psychological Health and Safety [6], and standards by the International Organization for Standardization (ISO) [7], that emphasize the importance of assessing multidimensional risks related to PF at work.

Numerous questionnaires and scales are available to measure and identify multiple PF at work and used in both research and practice. For example, in UK HSE management standards [5], the indicator tool helps assess employee perceptions of six key stressor areas: demands, control, support, relationships, role, and organizational change. The Copenhagen Psychosocial Questionnaire (COPSOQ) [8,9,10] measures a broad range of PF, including stressors, health and well-being, and personality. The third version of COPSOQ covers eight domains and 26 dimensions using validated items [10]. The Generic Job Stress Questionnaire (GJSQ) from the United States of America National Institute for Occupational Safety and Health (USA NIOSH) also covers various job stressors, mental health, and personality [11]. The Korean Occupational Stress Scale (KOSS) was developed in Korea; it consists of eight subscales of job stressors: physical environment, job demand, insufficient job control, interpersonal conflict, job insecurity, organizational system, lack of reward, and occupational climate [12]. Multidimensional scales to measure PF at work in specific industries were also developed and reported such as for construction workers [13], teachers [14], nurses [15], and dentists [16].

### 1.2. Background in Japan

In Japan, the 57 items of the Brief Job Stress Questionnaire (BJSQ) was developed in 2000 [17] based on the GJSQ from the USA NIOSH [11], covering job stressors, stress responses, buffering factors (i.e., social support), and job satisfaction. Among job stressors, the BJSQ includes quantitative job overload (three items), qualitative job overload (three items), physical demands (one item), job control (three items), skill utilization (one item), interpersonal conflict (three items), poor physical environment (one item), suitable jobs (one item), and meaningfulness of work (one item). Stress responses include vigor (three items), anger-irritability (three items), fatigue (three items), anxiety (three items), depression (six items), and physical complaints (11 items). Buffering factors include support from supervisors (three items), coworkers (three items), and family and friends (three items). Job and life satisfaction are also measured by a single item for each. All items are rated on a four-point scale. The New BJSQ was developed in 2014, covering effort–reward imbalance, bullying, organizational factors, work–self balance, and positive outcomes [14]. Most subscales in the BJSQ and the New BJSQ showed acceptable levels of internal consistency, test–retest reliability, and structural validity [18].

In the last two decades, the BJSQ has been used widely for occupational health studies and practice and tests the associations with a broad range of outcomes, including biological markers. The New BJSQ has also been used in later studies. Recently, the Japanese government launched a new occupational health policy called the National Stress Check Program (NSCP); this policy mandates that workplaces with 50 or more employees conduct assessments of psychosocial stress in employees at least once a year [19]. This policy recommends the use of the BJSQ as a structured questionnaire for the assessment. Thereafter, the BJSQ has been used more frequently, and the publication of data measured by the BJSQ has increased rapidly.

### 1.3. Research Gaps and Objectives

However, no systematic review has reported on the usage of the BJSQ and the New BJSQ and the findings of studies that used these questionnaires. For the COPSOQ, systematic reviews for the usage of the measurements have already been reported [20,21], and the international scientific production was summarized. Moreover, a systematic review of the BJSQ and the New BJSQ is important to make a milestone of scientific production from these measurements. Additionally, the summary of published data measured using the BJSQ and the New BJSQ, including samples, subscales, and scoring methods, would be useful statistics for research and practice in occupational health in Japan. The correlates of the BJSQ and the New BJSQ would be useful for validating the questionnaires and accumulating scientific evidence of the association between PF at work and health. This study aimed to systematically review observational studies that used the BJSQ and the New BJSQ to show their usability. Published literature until 2021 was systematically reviewed using various databases. The BJSQ subscales, scoring methods, and other validated measurements were qualitatively summarized. This study significantly contributes to creating a new summary of the questionnaires and encouraging the utilization of the questionnaires in future research and practice in occupational health.

## 2. Materials and Methods

### 2.1. Study Design

This study was a systematic review of observational studies. The reporting in this study was conducted following the updated guideline for reporting systematic reviews (the Preferred Reporting Items for Systematic Reviews and Meta-Analysis (PRISMA) 2020 statement) [22]. The study protocol was registered at the University Hospital Medical Information Network Clinical Trials Registry (UMIN-CTR, ID R000045091) in Japan.

### 2.2. Eligibility Criteria

For the systematic review, the authors included studies that (1) adopted observational study design; (2) sampled workers; (3) used at least one subscale of the BJSQ or the New BJSQ; (4) used other validated measurements and tested associations between the BJSQ or the New BJSQ subscales and other measurements; (5) were written in English or Japanese; and (6) were peer-reviewed. Studies were also included if they used the BJSQ and the New BJSQ subscales as non-primary variables and investigated the associations in preliminary analyses. Those studies that used other single-item measurements (e.g., smoking status), except for shift work, working hours, sleeping hours, subjective views of health, subjective well-being, and subjective satisfaction, were excluded.

### 2.3. Information Sources and Search Strategy

A systematic search of the literature on 13 September 2021 was conducted on databases such as MEDLINE (PubMed), EMBASE, PsycINFO/ARTICLES, and Japan Medical Abstract Society. For search terms, “brief job stress questionnaire” OR “BJSQ” was used, and no filter/limit was applied for any of the databases.

### 2.4. Study Selection and Data Collection Process

Identified records were managed in a Microsoft Excel (Washington, DC, USA) file. One investigator sorted the records by title and removed duplicates. Subsequently, each record was assigned to two reviewers from among 13 investigators. The investigators independently judged whether a record met the inclusion criteria of the systematic review. Records judged as not eligible by both of the two contributors were excluded, and other records were sought for retrieval of full texts. The full texts were judged by two independent reviewers, different from the initial screening, from 18 investigators. Reports assessed as eligible by both reviewers were included for review. When two investigators had inconsistent judgment at this full-text review stage, an agreement was reached through discussions with the project directors. When a report was excluded at this stage, the primary reasons for exclusion were recorded.

One of the reviewers of each study collected data from that study. The data were then reviewed by KW. The collected data included the names of the first authors, study design (cross-sectional or longitudinal), samples, subscales of the BJSQ and the New BJSQ, scoring methods of the BJSQ and the New BJSQ, and other validated measurements.

### 2.5. Data Synthesis and Analysis

Since this study aimed to summarize the usage of the BJSQ and the New BJSQ, no statistical data synthesis was conducted. Assessments of risk of bias within individual studies, heterogeneity, reporting bias, and certainty of evidence were not required to be conducted either. The collected data in the text and tabulation were qualitatively summarized. In addition, the number of subscales of the BJSQ and the New BJSQ and other measurements used were counted and visually placed, classifying them into five categories according to the job stress model [11]: (1) job stressors or exposures that relate to work conditions which lead to stress responses; (2) health-related outcomes including physiological and psychological responses; (3) work-related outcomes such as job satisfaction, job performance, and burnout; (4) individual and behavioral factors that modify the associations between job stressors and outcomes; and (5) buffering and non-work factors such as social support.

## 3. Results

### 3.1. Study Selection

Figure 1 illustrates the selection process of this systematic review. A systematic search of databases resulted in 741 hits. After the initial screening and the full-text review by the independent reviewers, 145 reports published from 2003 to 2021 were included in this systematic review [23,24,25,26,27,28,29,30,31,32,33,34,35,36,37,38,39,40,41,42,43,44,45,46,47,48,49,50,51,52,53,54,55,56,57,58,59,60,61,62,63,64,65,66,67,68,69,70,71,72,73,74,75,76,77,78,79,80,81,82,83,84,85,86,87,88,89,90,91,92,93,94,95,96,97,98,99,100,101,102,103,104,105,106,107,108,109,110,111,112,113,114,115,116,117,118,119,120,121,122,123,124,125,126,127,128,129,130,131,132,133,134,135,136,137,138,139,140,141,142,143,144,145,146,147,148,149,150,151,152,153,154,155,156,157,158,159,160,161,162,163,164,165,166,167]. Of the included reports, 102 had digital object identifiers, and the main text of 52 reports was written in Japanese.

A total of 230 reports were excluded at the full-text review stage, although some of them might have met the inclusion criteria. For example, Eguchi et al. [168] investigated the association between psychological stress response measured by the BJSQ and workplace occupational mental health (OMH) and related activities. However, the items of OHM activities were derived from a paper by the Japanese government and were not psychometrically validated. Kawada and Otsuka [169] conducted a longitudinal study to examine changes in job stress and job satisfaction using the BJSQ. However, they only reported the associations among the subscales of the BJSQ, not with other validated measurements. Iguchi [170] examined the associations among job demands, job resources, and turnover intention among public health nurses using the BJSQ and the New BJSQ. However, this study conducted a factor analysis for the subscales and conceptualized new variables in the analysis. Some studies used the BJSQ overseas: China, India, and the USA [171,172,173]. These studies did not report the validity of the translated version of the BJSQ.

### 3.2. Study Characteristics

A summary of the included studies is shown in the Table A1. Most studies were conducted cross-sectionally (n = 116) [52,53,54,55,56,57,58,59,60,61,62,63,64,65,66,67,68,69,70,71,72,73,74,75,76,77,78,79,80,81,82,83,84,85,86,87,88,89,90,91,92,93,94,95,96,97,98,99,100,101,102,103,104,105,106,107,108,109,110,111,112,113,114,115,116,117,118,119,120,121,122,123,124,125,126,127,128,129,130,131,132,133,134,135,136,137,138,139,140,141,142,143,144,145,146,147,148,149,150,151,152,153,154,155,156,157,158,159,160,161,162,163,164,165,166,167], while the remaining were longitudinal studies (n = 29) [23,24,25,26,27,28,29,30,31,32,33,34,35,36,37,38,39,40,41,42,43,44,45,46,47,48,49,50,51]. The sample size ranged from 18 [83] to 69,805 [60]. In the included studies, recruitment of the participants was conducted from private companies (n = 59), hospitals (n = 42), nursing or welfare facilities (n = 13), healthcare centers (n = 7), web surveys (n = 5), public sectors (n = 5), existing cohorts (n = 4), fire defense stations/headquarters (n = 4), and a convenience sample of faculty staff members or alumni of universities (n = 6).

### 3.3. Used Subscales and Other Measurements

Figure 2 shows the list of used subscales from the BJSQ, the New BJSQ, and other measurements in the included studies. Parenthesis in each subscale shows the number of times the measurements have been used.

For the subscales of the BJSQ, job stressors (n = 95), especially quantitative job overload (n = 65) and job control (n = 64), were most often used; stress responses (n = 88) and social support (n = 72) were frequently used as well. Most of the studies referred to the job stress model from the US NIOSH [11] or the job demands–control model by Karasek [174,175]. For example, Izawa et al. [95] used the subscales of quantitative job overload and job control from the BJSQ, calculated the job strain index by dividing quantitative job overload by job control, and investigated an association with cortisol levels in fingernails. Hidaka et al. [54] also adopted job strain through quantitative job overload and job control and social support from supervisors and coworkers of the BJSQ. They indicated those significant associations with health-related quality of life among Japanese workers. Stress responses were often used as the health outcomes explained by PF at work. A two-year follow-up study by Taniguchi et al. [37] investigated the association between workplace bullying and harassment and stress responses of the BJSQ among care workers at welfare facilities for the elderly. They reported multiple types of bullying and harassment were positively associated with psychological stress response at the follow-up. Shimazu and de Jonge [48] also used stress responses from the BJSQ as an outcome of the effort–reward imbalance and reported the reciprocal associations in a three-wave panel survey. Satisfaction (n = 27) was mainly utilized for examining associations with other health outcomes. Inoue et al. [28] investigated the prospective association between job satisfaction of the BJSQ and long-term sickness absence. They indicated that workers who perceived job dissatisfaction had a significantly higher risk of long-term sickness absence; however, after additionally adjusting for the psychosocial work environment, this association was weakened and was no longer significant.

The subscales of the BJSQ were utilized for investigating the relationships among various kinds of other measurements: 13 job stressors and exposures, 28 health-related outcomes, 14 work-related outcomes, 19 individual and behavioral factors, and three buffering and non-work factors. For health outcomes, the most often used measurement was depression and anxiety (n = 17). For instance, Tsuboi et al. [51] investigated the association between job stressors and depressive symptoms measured by the center for epidemiologic studies with a depression scale. They compared and categorized female nurses into the most stressful group and the least stressful group and reported a significant difference in depressive symptoms between the two groups. Sakamoto et al. [75] investigated the structural differences among factors for psychological job stress among healthcare workers and reported that job stressors from the BJSQ were positively associated with depression and anxiety measured by the hospital anxiety depression scale. In addition, sleep/insomnia/circadian rhythm (n = 11) were frequently investigated for their association with job stressors. Toyoshima et al. [55] examined interrelationships among sleep reactivity, job-related stress, and subjective cognitive dysfunction and indicated that sleep reactivity significantly influenced subjective cognitive dysfunction directly and indirectly via job stressors and stress responses. Takahashi et al. [46] conducted a one-year longitudinal study to examine how a change in work time control was associated with sleep and health. They indicated that daytime sleepiness was positively associated with quantitative job overload and negatively associated with job control and social support from the BJSQ.

Physiological health outcomes were also tested using the subscales of the BJSQ, such as diabetes, insulin resistance, and blood glucose (n = 4), serum lipid and cholesterols (n = 4), salivary or fingernails cortisol (n = 2), and inflammatory markers (n = 1). For example, Sugito et al. [23] conducted a retrospective study with male workers to investigate the effects of job stressors on the onset of diabetes mellitus defined by HbA1c or using antidiabetic drugs. They indicated that low skill utilization from the BJSQ was associated with the risk of diabetes mellitus onset. Watanabe et al. [101] examined interrelationships between job resources, vigor, exercise habit, and serum lipids including triglyceride, high-density lipoprotein cholesterol, and low-density lipoprotein cholesterol. Multiple-group path analysis indicated that job resources and vigor from the BJSQ were inversely associated with triglyceride and low-density lipoprotein cholesterol and positively associated with high-density lipoprotein cholesterol through exercise habits in both sexes. Nakata et al. [121] investigated associations between job stressors and inflammatory markers including high-sensitive C-reactive protein, interleukin-6, tumor necrosis factor-alpha, monocyte, and leukocyte. The job strain index calculated by dividing quantitative job overload by job control was negatively associated with tumor necrosis factor-alpha.

For work-related outcomes, burnout (n = 7) and presenteeism (n = 7) were often investigated as those associations with PF at work. Saijo et al. [106] investigated the synergistic interaction of job demands, job control, and social support on mental health among local government employees. They indicated significant associations between these stressors from the BJSQ and burnout measured by the Japanese version of the Maslach Burnout Inventory-General Survey. Hayashida et al. [57] assessed the association between the irregularity of mealtimes and presenteeism measured by the Work Limitations Questionnaire. They indicated that the irregularity of mealtimes had a strong effect on presenteeism indirectly through psychological and physical stress responses from the BJSQ.

Coping (n = 10) and sense of coherence (n = 6) were frequently used as individual and behavioral factors. For instance, Shimazu et al. [49] examined the lagged effects of active coping on stress responses to explain the individual differences in the underlying mechanisms behind the association between job stressors and health outcomes. They measured active coping using the Brief Stress for Coping Scale and reported significant interactions of quantitative job overload, job control, and active coping on stress responses from the BJSQ. Urakawa et al. [45] examined the association between a sense of coherence and psychological responses and reported that a sense of coherence was inversely associated with psychological and physical stress responses.

A total of five reports used subscales from the New BJSQ [30,36,81,94,111]. Morimoto et al. [30] investigated the adverse effects of role conflict on the psychological strain among employed family caregivers of people with dementia. They used the subscales of emotional demands and role conflict from the New BJSQ, in addition to the subscales from the BJSQ. They indicated that conflict between caregiving and work was positively associated with psychological strain and its association was moderated by formal support seeking and attentional control. Sakuraya et al. [36] used a three-item scale of workplace social capital and investigated its association with the onset of major depressive episodes through a three-year prospective cohort study. The study indicated that middle-level workplace social capital had the lowest risk of major depressive episodes. Inaba [81] and Inaba and Inoue [111] used multiple subscales from the short version of the New BJSQ and examined their associations with subjective well-being and burnout among female nurses. They indicated that burnout was significantly associated with role conflict, role clarity, and job security [111], and subjective well-being was significantly associated with career development [81]. Toyama and Mauno [94] used the three-item subscale of realization of creativity and reported a significant and positive association with emotional intelligence among eldercare nurses in special nursing homes.

### 3.4. Scoring Methods

Concerning scoring methods, most studies used continuous scores of the subscales (n = 111). Categorization using means, medians, tertiles, and quantiles was also adopted (n = 12). Standardized scores on a five-point scale (n = 7) were calculated based on the distribution of continuous scores in the representative sample [176]. For more practical and easier scoring, a simple scoring method was used (n = 9) [177]. In this method, the respondents were dichotomized into stressed or not stressed, by counting how many items of the BJSQ were scored as undesirable. The definition of “high-stress” employees according to the Japanese NSCP was also used (n = 7) [32]. This definition is conceptualized by the combination of high scores in stress response, high scores in job stressors, and low scores in social support. The predictive validity of the “high-stress” employees for long-term sickness absence at the one-year follow-up was confirmed in a previous study [32].

## 4. Discussion

### 4.1. Main Findings

In the last two decades, over 140 observational studies using the BJSQ and/or the New BJSQ have been published. Since 2015, when the NSCP was launched, large-scale data from more than 60,000 people have been published, as the assessment of psychosocial stress in employees became mandatory. Although not all reports were written in English, more than two-thirds were readable, at least with abstracts that were in English, and more than 100 articles were identifiable by digital object identifiers. Associations were established between a wide variety of factors, including job stressors, health-related outcomes, work-related outcomes, individual and behavioral factors, and buffering factors. The relationship with physical biomarkers was also examined. Although not all studies observed significant associations between factors, and not all study hypotheses were supported, the reported associations were generally reasonable and consistent with existing findings about job stress models. This means that the mechanism that exposure to job stressors evokes deterioration of health- and work-related outcomes and that some of these associations are modified by individual and behavioral factors. Therefore, the BJSQ and the New BJSQ are questionnaires that have made substantial contributions to the research and practice of occupational stress in Japan.

### 4.2. Theoretical Implications

The reasonable associations with validated measurements of health- and work-related outcomes were repeatedly observed in multiple subscales of the BJSQ. In particular, quantitative job overload, job control, supervisor and coworker support, and stress responses often had significant associations with depression and anxiety, quality of life, sleepiness, burnout, sickness absence, and physical biomarkers. These results may reflect the construct validity (concurrent and predictive) of the subscales, while the BJSQ is easy to answer because of the low number of items in each subscale (three at most). The subscales of quantitative job overload, job control, and social support at work can be used as the representative job stressors, referring to the job demands–control model [174,175] as the theoretical background. The subscales of the psychological and physical stress response may also be useful as the indicators of broad symptoms evoked by exposure to stressful PF at work. These subscales may be used as the outcomes of the intervention study. In contrast, compared to the subscales from the BJSQ, those from the New BJSQ were not much considered in the research. More studies are needed to confirm the psychometric validity using the subscales from the New BJSQ.

### 4.3. Practical Implications

Scoring methods are inconsistent among studies, which is partly because these were developed so that the BJSQ can be used for both research and practice. The predictive validity of sickness absence has been confirmed for “high-stress” employees in the NSCP. It is necessary to use the appropriate method according to the purpose of use.

Translation into other languages is the next interest in research and practice. Several studies were conducted in other countries but were not included because the validity of the translated scales could not be verified [171,172,173]. Recently, the BJSQ and the New BJSQ have been translated into English, Chinese, Portuguese, Myanmar, Vietnamese, Spanish, Tagalog, Nepali, Persian, and Indonesian [178,179], and the use of these scales in other countries have already been reported in a peer-reviewed journal [180]. The translated version of the scales can be used as tools to promote not only research on foreign workers in each region of Japan but also job stress research in other countries and international job stress research.

### 4.4. Limitations

There are several limitations to this study. The study quality and risk of bias of the included studies were not assessed because the objective of this study was limited to summarizing published information related to the BJSQ and the New BJSQ. Since most of the included studies were conducted cross-sectionally, the findings from each study could include substantial biases. Further, a body of evidence for the associations between the subscales from the BJSQ and other measurements could not be presented.

## 5. Conclusions

In conclusion, as a comprehensive questionnaire, the BJSQ and the New BJSQ have contributed to the measurement of PF at work and the publication of scientific papers in the occupational health field. The BJSQ can be one of the methodological tools to explore the mechanisms between job stress and several work-related disorders and can provide hints of intervention. Quantitative job overload, job control, and supervisor and coworker support were often used and may have the construct validity as the representative job stressors, referring to the job demands–control model. Regarding practical implication, using the appropriate scoring method according to the usage purpose is important. Prospective, interventional, and multilingual studies are expected to be published to accumulate more comprehensive and high-quality findings in the future.

## Figures and Tables

**Figure 1 ijerph-20-01814-f001:**
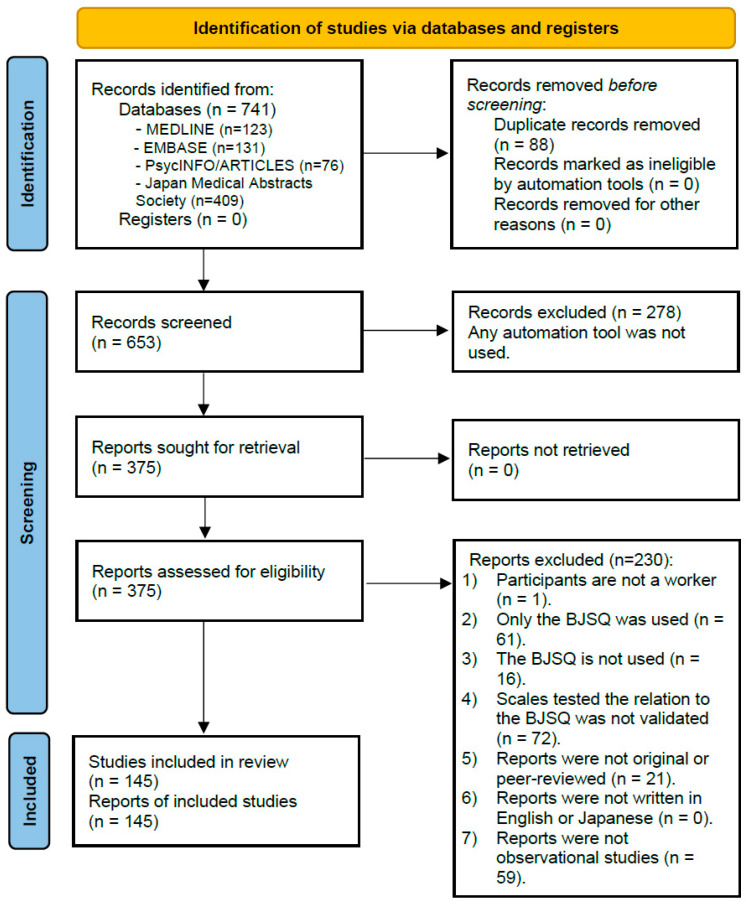
PRISMA 2020 flow diagram for new systematic reviews which included searches of databases and registers only.

**Figure 2 ijerph-20-01814-f002:**
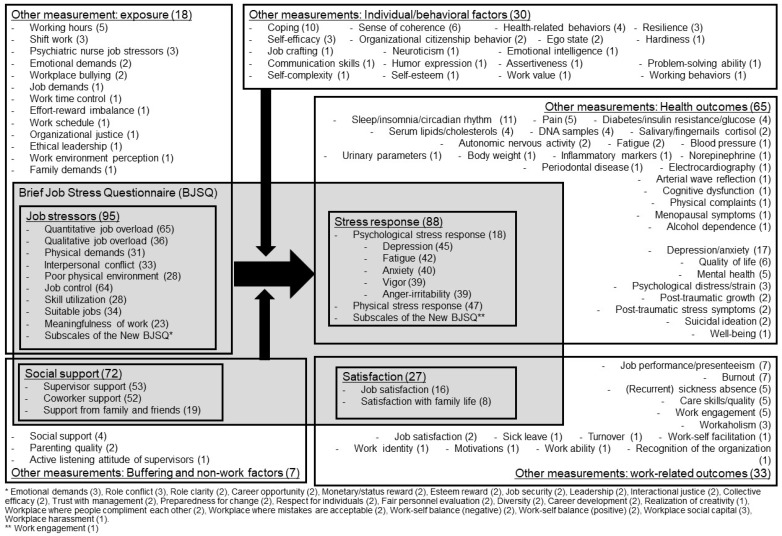
Used subscales from the BJSQ, the New BJSQ, and other measurements in the included studies. Note. Parenthesis in each subscale shows the number of times the measurements have been used.

## Data Availability

Since this study is a systematic review, no individual data is available. The summarized data for this systematic review can be obtained upon request.

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
