# Peer review of "Usage of the Brief Job Stress Questionnaire: A Systematic Review of a Comprehensive Job Stress Questionnaire in Japan from 2003 to 2021"

_ijerph, 2023, doi:10.3390/ijerph20031814_

Round 1
Reviewer 1 Report
Please check the reviewer's comment on the attached file.

Author Response
Please see an uploaded Word file.

Reviewer 2 Report
Usage of the Brief Job Stress Questionnaire: a systematic re- view of a comprehensive job stress questionnaire
The idea of the study seems interesting, different and even necessary
Abstract
The abstract is fine, but unattractive, I suggest that to follow the solid scientific work structure and be restructured in a way that includes the following clearly:
· The contextualization of the study
· The main objective
· The justification
· The sample used
· The methods used
· The main findings and conclusions
· The novel contribution
Introduction and Literature Review
1.The introduction needs to be more clear and straight to the point by justifying soundly on the main objective; a systematic re- view of a comprehensive job stress questionnaire
2.The authors must concentrate of the importance of this kind of re-view.
3. State the method at the end of the introduction, as well as the study's novel contributions.
4. Literature section needs to be improved. Not enough literature has been provided. The sources cited are not enough. The research gaps in the previous studies related to this topic were not explained clearly. Please address these issues.
5. I suggest that, the Literature section needs to be added as a separate headline. While establishing the hypotheses or the questions, the authors must give an extensive background.
Methodology.
fine.
Results.
fine
Discussions
1. Some aspects of the discussion are included in the results section.
2. Theoretical, practical implications and limitations should be transferred to conclusion section or alone, and more clear.
The conclusions
Should be improved, including a clear theoretical implication and practical implication of the research.
Others:
The paper has some editing issues. It needs proofreading.
Good luck
Author Response
Please see an uploaded Word file.

Round 2
Reviewer 1 Report
Good Job!!